# Physiochemical and Sensory Properties of Bread Fortified with Wheat Bran and Whey Protein Isolates

**DOI:** 10.3390/foods12132635

**Published:** 2023-07-07

**Authors:** Jaromír Pořízka, Zuzana Slavíková, Karolína Bidmonová, Miroslava Vymětalová, Pavel Diviš

**Affiliations:** 1Faculty of Chemistry, Brno University of Technology, 612 00 Brno, Czech Republic; zuzana.slavikova@vut.cz (Z.S.); divis@fch.vut.cz (P.D.); 2Mlýny J. Voženílek, Ltd., Průmyslová 107, 503 02 Předměřice nad Labem, Czech Republic; pvj@mlynyvozenilek.cz

**Keywords:** high protein food, pastry, wheat bran, protein isolates, whey protein, rheology, sensory analysis

## Abstract

This study investigated the effect of fortifying baked goods with wheat bran (WBPI) and whey protein isolates (WPI) on their physicochemical and sensory properties. The aim was to enhance the nutritional value by incorporating high-protein ingredients. WBPI and WPI, which are rich in essential amino acids, were chosen to create high-protein flour blends. The main advantage of WBPI is that it is derived from readily available and inexpensive wheat bran. High-protein flour blends fortified with substitutions of 5%, 10%, and 15% flour with WBPI and WPI were subjected to chemical and rheological analysis. WBPI substitution slightly increased water binding and softening, but it resulted in a decrease in dough quality. In contrast, WPI substitution prolonged dough development time, improved dough stability, and enhanced farinographic quality. WBPI-substituted dough exhibited comparable extensographic properties to the reference flour, with 5% WBPI substitution leading to improved energy and dough resistance. However, as the level of WBPI flour substitution increased, extensographic parameters gradually declined without further enhancing the dough’s mechanical properties. Samples with 5% WPI substitution demonstrated superior mechanical properties compared to the reference sample. Baguette with high WBPI substitution was associated with reduced overall acceptance due to a bitter taste caused by the presence of small peptides, ferulic acid, and tannins, as confirmed by correlation analysis.

## 1. Introduction

Pastries are a popular food worldwide and are a regular part of many people’s diets. Currently there is growing interest in creating pastries with higher nutritional value and fewer calories. This has led researchers and food manufacturers to explore the potential of fortifying pastries with functional ingredients that can provide additional nutritional benefits. Increasing the protein content by using protein isolates is currently one of the most promising approaches for the production of modified food [1,2,3,4,5,6,7].

Protein additives in general can be used as functional ingredients in preparing high-protein foods and can be of both animal and plant origin. Physiochemical properties that can be affected by the addition of protein isolates, include texture, color, and nutrition. For example, the addition of proteins to pastry formulations can lead to changes in rheology and specific volume, which may result in changes in the texture and mouthfeel. Additionally, the polyphenols present in plant isolates can cause the finished product to have a darker color. Plant protein isolates in pastries can also increase the nutritional value by increasing protein and fiber while reducing fat [8,9,10].

One isolate with a high potential for use in the baking industry is wheat bran protein isolate (WBPI). It is isolated from the outer layer of the wheat grain through a series of processing steps that include defatting, solubilization, and precipitation. The essential advantage of WBPI is that it is produced from commonly available wheat bran, which is a cheap and sustainable material produced in large quantity during milling [11].

WBPI is a plant-based protein isolate with a well-balanced amino acid profile that is comparable to plant proteins like soy and pea and has a very similar Protein Digestibility Corrected Amino Acid Score (PDCAAS). The isolate is particularly rich in asparagine, glutamic acid, leucine, arginine, and proline. In addition to its protein content, WBPI also contains bioactive compounds such as polyphenols, phytosterols, and lignans that have been linked with health benefits, such as improved cardiovascular health, antioxidant activity, and anticancer properties [12].

The food application of WBPI has not been extensively studied although a few studies have described its functional properties in model systems. It has been proven that WBPI has the potential for application in food formulation due to its good emulsifying and foaming properties and high water and fat holding capacity [13]. WBPI has also been experimentally used as a protein fortifier in pasta [14] and bread [15]. Spaghetti quality was significantly affected when 5–10% of WBPI was used to enhance the protein content. The deterioration of functional properties was observed with higher WBPI substitutions, but cooking loss and appearance were still comparable to common whole-meal pasta. The evaluation of the influence of WBPI fortification on the functional properties of bread is more complicated. Alzuwaid et al. [14] and Uttam et al. [15] achieved different results when assessing the effect of fortification on loaf volume. From a detailed study of the results, a substantial improvement in volume was achieved when only the albumin fraction of WBPI was used. Non-fractionated WBPI had the opposite effect.

Based on previously published findings, it is evident that the effect of fortifying pastries with WBPI on physiochemical and sensory properties is still not sufficiently understood. The aim of this study is to expand the available knowledge with new findings by using conventionally used analytical techniques from the field of dough analysis such as farinography and extensography. A baking experiment was also part of the study. High-protein baguettes were prepared and subjected to a comprehensive sensory analysis. This study also includes a direct comparison of WBPI and whey protein isolate (WPI) as common representatives of animal proteins to explore the advantages and disadvantages of both functional ingredients.

## 2. Materials and Methods

### 2.1. Protein Isolates and Analysis

The WBPI was obtained from wheat bran (Mlýny J. Voženílek, Předměřice nad Labem, Czech Republic) using the pH shift method with modifications (Figure 1). The protein fraction was extracted using a NaOH (Lach-Ner, Neratovice, Czech Republic) solution (pH 10.5 ± 0.1) in a ratio of 1:20 (wheat bran/NaOH solution) for 2 h with continuous stirring. The liquid protein extract was separated from deproteined wheat bran by centrifugation (8000 rcf, 15 min, room temperature) and its pH was adjusted to 4.0 ± 0.05 using 1 M citric acid (Lach-Ner, Neratovice, Czech Republic). The precipitated WBPI was collected after centrifugation (8000 rpm, 15 min, room temperature), lyophilized, and used for further experiments.

The WPI used for the experiments was a commercial whey protein (Vilgain, Brno, Czech Republic).

The protein content and amino acid profile of WBPI were determined. The amount of nitrogen was analyzed using Eurovector EA3100, and the protein was obtained by multiplying the nitrogen content by the conversion factor 6.31, which is specific to wheat-bran proteins. The amino acid profile was determined by HPLC. A 10 mg sample of protein material was hydrolyzed in 3 mL of 6 M HCl for 12 h at 110 °C. The solution was dried and reconstituted in 5 mL of water. HPLC analysis was performed according to the method described by Gorissen et al. [16]. A single quadrupole LC/MSD iQ Agilent 1260 (Agilent, Santa Clara, CA, USA) was used with AdvanceBio Amino Acid Analysis (AAA), 3.0 × 100 mm column (Agilent, USA).

### 2.2. Preparation and Characterization of Flour Samples

Plain high-gluten wheat flour T530 (Mlýny J. Voženílek, Czech Republic) was selected for the preparation of high-protein flour blends and for all baking experiments. Fortification of flour was performed by the direct mass substitution of part of the flour with either WBPI or WPI. Composition of flour mixtures with protein isolates is presented in Table 1.

#### 2.2.1. Determination of Wet Gluten

Exactly 10 g of the flour mixtures were weighed into a porcelain bowl. A 2% sodium chloride solution was added to the protein isolate in a volume of 5–5.5 mL. Using a spatula, a moderately stiff dough was prepared and then kneaded well and shaped into a small ball. The dough covered by a watch glass and left for 30 min. Afterwards, the dough was washed under a gentle stream of water using a sieve. The washing was stopped when water droplets expelled from the wet gluten were transparent. This measurement was performed on samples of flour, flour with WBPI substitution (5, 10, and 15%), and flour with wheat protein substitution (5, 10, and 15%).

The content of wet gluten, expressed as a weight percentage of the product’s dry matter, was calculated using the following equation: $$X=m\cdot 10\cdot \frac{100}{100-w_{1}},$$
where *m* is the weight of washed dough, and *w_1_* is the water content (%).

#### 2.2.2. Determination of Sedimentation Value

Exactly 80 mL of lactic acid was transferred into a sedimentation flask, which was then brought to a temperature of 27 °C. From freshly washed gluten, 1 g was weighed and torn into approximately 30 equal pieces. In a porcelain dish, 10 mL of tempered lactic acid was added along with the small pieces of gluten. The content of the porcelain dish was transferred to the swelling flask containing the main portion of the lactic acid solution. The flask was heated in a water bath to 27 °C and left for 120 min. After 10 and 20 min, the contents of the flask were gently mixed to prevent the gluten pieces from sticking together or adhering to the bottom. The flask was stoppered in such a way that the stopper reached the zero mark on the scale. By gradually tilting the flask, the swollen gluten particles slid into the calibrated neck. The volume was immediately read on the scale to the nearest 0.5 division. This measurement was performed on gluten from T530 flour, gluten with WBPI substitution (5, 10, and 15%) and gluten from flour with whey protein substitution (5, 10, and 15%).

#### 2.2.3. Water and Elemental Analysis of Flour Samples

The water content in all flour samples was analyzed using the Karl Fisher method developed by Laura Corpaş et al. (2013) [17]. Karl Fisher titration was performed on KF Titrando (Metrohm, Herisau, Switzerland). Elemental analysis of C, H, N, S was performed by the elemental analyzer Eurovector EA3100 (Eurovector, Pavia, Italy).

#### 2.2.4. Farinographic Analysis of Flour Samples

The farinographic analysis was carried out at the milling company Mlýny J. Voženílek in Předměřice nad Labem (Czech Republic) on Farinograph-TS instrument (Brabender, Duisburg, Germany). The main component of the farinograph was a mixer with a capacity of 300 g equipped with blades rotating at a speed of 63 revolutions per minute. It also had a thermostat that ensured a constant temperature of 30 °C during the experiment. Exactly 300 g of flour were added to the mixer, and an appropriate water volume was introduced using a burette. The water was added until the torque value reached 500 farinographic units (FUs), which is standard for flour testing in industrial milling. Farinographic analysis was performed on samples of T530 flour, T530 + 5% WBPI, and T530 + 5% WPI. Higher percentage concentrations of WPI and WBPI were not tested because the analytical limit of 500 FUs could not be reached with this type of instrument. Farinographic indicators were calculated:$$\mathrm{Water}\;\mathrm{absorption}\;=\;\frac{m_{water}}{m_{flour}}\;\times \;100$$

The dough-development time indicated the duration required for dough to reach 500 FUs. The stability of dough refers to its ability to maintain its structure and resist weakening or breakdown during mixing while the torque remains at 500 FUs. The degree of softening indicated a decrease in dough resistance to mixing represented as an FU after 10 and 12 min. The farinographic quality number (FQN) indicated the time from the beginning of measurement to the 30 FU drop from the maximum curve.

#### 2.2.5. Extensographic Analysis of Flour Samples

The extensographic analysis was performed on a Extensograph-E instrument (Brabender, Germany) heated to 30 °C. The instrument settings are listed in Table 2. The extensographic measurement was conducted on T530 flour, T530 flour with 5, 10, and 15% substitution of flour with WBPI and WPI.

Dough was prepared using a mixer, which was part of the farinograph. Additionally, 6 g of sodium chloride were weighed and dissolved in the appropriate amount of water to prepare 0.1 M solution. This solution was used to prepare the dough of 400 FU consistency. The optimized amount of water for each mixture is provided in Table 3.

The dough was formed into a small ball using a shaper and then placed into a sheeter and reshaped into a cylinder. Dough cylinders were placed into a chamber heated to 30 °C and left to rest for 30 min. After the resting period, the dough was stretched by the device’s hook until it broke after which it was kneaded again, reshaped in the sheeter, and allowed to rest for another 30 min. Experiment was repeated. The resistance exerted by the hook on the dough during stretching was recorded.

#### 2.2.6. Baking Experiments and Sensory Analysis of Pastry

Dough for the baking experiment was prepared by mixing of flour blends, salt, yeasts and water. The ratio of the raw materials is described in Table 4. The pastry was shaped into baguettes and baked at 240 °C for 20 min. Sensory analysis made from reference and fortified flour was assessed by 20 evaluators in a specialized laboratory. Textural and flavor parameters were also evaluated.

#### 2.2.7. Statistical Analysis

Hypothesis testing was used to evaluate the impact of WBPI- and WPI-fortified flour on the physiochemical and sensory properties of dough and pastry. The dataset was processed using the Statistica software (TIBCO, Palo Alto, CA, USA, version 13.0). ANOVA and Kruskal–Wallis ANOVA was used to assess statistically significant differences among the fortified bakery products. Hypothesis testing was conducted at a significance level of α = 0.05.

## 3. Results

### 3.1. Analysis of WBPI and WPI

An analysis of WBPI and WPI revealed interesting amino acid profiles, which are presented in Table 5 along with the protein content. Both protein isolates are rich in amino acids that are important for sports nutrition. WPI is particularly rich in leucine, lysine, isoleucine, and valine. These branched-chain amino acids (BCAAs) make up approximately 70% of the amino acids in muscles. BCAAs are recognized for their role in promoting muscle growth and facilitating recovery. Previous studies have shown that they can increase the rate of protein synthesis while reducing protein degradation during periods of rest. Consequently, athletes commonly use these amino acids as dietary supplements, which are often incorporated into mixtures referred to as BCAAs. As a result, these products have gained popularity in the range of dietary supplements available for athletes. WPI also contains significant amounts of threonine, and among the non-essential amino acids, glutamine and aspartic acid are the most abundant [18].

WBPI also contains a range of nutritionally important amino acids, and like whey protein it has a significant amount of essential BCAAs. The most abundant non-essential amino acid in WBPI is aspartic acid, followed by glutamine and arginine, which plays a role in regulating urea synthesis in the liver and stimulating the secretion of growth hormone and insulin. Growth hormone is important in the post-physical performance recovery phase [19].

During storage, the lysine in wheat flour degrades. Products made from raw materials stored for a longer period may have a lower concentration of lysine. This deficiency can be solved by substituting a portion of the flour with a protein isolate. Since both WBPI and WPI are rich in lysine, fortifying bakery products could supplement this limiting amino acid [19].

WPI is known for its high PDCAAS score of 1, whereas WBPI, a plant-based protein, has a lower score of 0.54 with phenylalanine being the limiting amino acid [20].

### 3.2. Analysis of Flour Samples

The protein, water, and wet gluten content and the gluten sedimentation volume of the reference flour and the high-protein blends from WPI and WBPI substitution were determined. All samples were prepared following the procedures described in Section 2.2. The results of the analysis are presented in Table 6.

### 3.3. Rheological Properties of Prepared Doughs

#### 3.3.1. Farinographic Analysis of Dough

The purpose of the farinographic measurement was to assess the impact of WBPI and WPI substitution on the rheological properties of the dough following the procedure described in Section 2.2.4. Table 7 displays the values of individual farinographic indicators for all samples: water absorption, development time, stability, softening degree, and farinographic number. Two softening degrees were recorded: 10 min as measured from the start of the measurement and 12 min as measured after reaching maximum torque.

A reference sample of T530 flour without any protein substitution was used. One important physical characteristic of wheat flour is its water absorption capacity. Strong flours typically exhibit a water absorption range of 55–60%, meaning that 100 g of wheat flour can absorb 55 to 60 g of water [21]. The analyzed T530 flour fell within this range, indicating its strength and suitability for bread baking due to its high gluten content. Dough development time and dough stability indicate flour strength. The higher values indicate a stronger dough where wheat gluten plays a crucial role in forming a three-dimensional viscoelastic structure. In the Czech Republic the parameter of dough stability typically ranges from 3 to 5 min for wheat flours. The reference T530 flour, based on the measurement, achieved a stability value of 3.8 min.

Data from the farinograph (Table 8) demonstrated that replacing wheat flour with WBPI resulted in a slight increase in water binding compared to the reference sample. This increase could be attributed to the higher protein content and the presence of fiber, which contains numerous hydroxyl groups that interact with hydrogen bonds in water, thereby enhancing absorption. This finding is consistent with the results of a study conducted by Manuel Gómez et al. [22] that investigated the effect of extruded wheat bran on dough rheology and bread quality. Substituting flour with WBPI did not significantly alter the dough development time: only by 0.1 min compared to the reference sample. During dough development, interactions between fiber and gluten can occur, potentially prolonging the process. However, the WBPI, despite its fiber content, did not lead to this phenomenon [22].

On the other hand, dough stability exhibited a noticeable decrease. The substitution of 5% WBPI resulted in a weakening of the gluten network and a decrease in flour strength. Consequently, stability decreased by 1.2 min compared to the reference sample, a significant difference. Additionally, WBPI substitution led to a slowdown in dough softening, which is inversely related to the farinograph quality number (FQN).

Farinographic analysis was also conducted on flour blends with WPI. The data from the farinograph (Table 8) indicated a decrease in water absorption compared to the reference sample. Substituting flour with WPI resulted in the formation of a complex system that hindered hydration, extensibility, and wheat gluten alignment, leading to an increase in dough development time by 5 min with 5% WPI. In addition, dough stability increased by 6 min compared to the reference sample. These findings align with a study conducted by Indrani et al. [23], which demonstrated that the substitution of up to 10% whey protein enhances dough stability. However, a different conclusion was reached in the study published by Tang et al. [24], who found that the substitution of WPI caused a decrease that was believed to be caused by interactions between the sulfhydryl group in WPI and the disulfide bond in wheat gluten. In their study, the substitution of 10% WPI to the dough resulted in a stability decrease: stability time value, 6.5 min. These contrasting results highlighted the fact that the substitution of WPI to different types of flour had varied effects on rheological properties, suggesting a need for further study.

The softening degree, which is determined by the difference between the consistency value at the maximum and the consistency value at 10 min, decreased due to longer dough development time. This implied that as the development time approached 10 min, the time for the curve to drop became shorter. It can be assumed that the softening time after 12 min will increase with a higher amount of whey protein. According to the FQN, which increased approximately threefold compared to the reference sample, the dough quality improved with a 5% whey protein substitution.

#### 3.3.2. Extensographic Analysis of Dough

The aim of the extensograph measurement was to determine the influence of the WBPI and WPI substitution on the rheological properties of the dough, according to the procedure described in Section 2.2.5. The values of individual extensographic indicators––energy, resistance, extensibility, and ratio number––are listed in Table 8. Two ratio numbers were recorded during the measurement. The ratio number was between resistance and energy, and the ratio number (max) was defined as the ratio between the maximum and energy.

Problems with extensive adhesiveness of the dough caused poor formation of fillet and roller for the 10 and 15% of WPI substitutions. The dough stuck to the cylinder and rolling pin, making it impossible to complete the extensograph determination. The adhesiveness of the dough could potentially have been eliminated by reducing the water content, but for both farinographic and extensographic determination, it was always necessary to create a dough of maximum consistency, reaching approximately 500 FUs. Within the extensograph determination, only the sample with 5% WPI substitution was analyzed.

**Table 8 foods-12-02635-t008:** Extensographic parameters of T530 flour and high protein blends.

	T530		5% WBPI		10% WBPI		15% WBPI		5% WPI	
Resting time (min)	30	60	30	60	30	60	30	60	30	60
Energy (cm^2^)	82	82	81	97	56	55	44	42	119	164
Resistance (FU)	216	230	728	960	524	276	144	64	289	196
Ductility (mm)	185	176	85	80	58	54	53	51	196	188
Max (FU)	317	343	741	961	799	865	680	691	443	657
Ratio number	1.2	1.3	8.6	12.0	9.0	4.6	2.7	2.1	1.5	2.2
Ratio number (max)	1.7	2.0	8.8	12.0	13.9	16.1	12.9	13.6	2.3	3.5

The energy of the dough was considered a measure of its bakery workability and dough quality. The lower the energy, the less resistant and stable the dough during processing. Flours with a rigid and short gluten structure typically have low dough energy. The measured energy value for the reference sample was 82 cm^2^, which corresponded to the standard for Czech flours, where this parameter is around 90 cm^2^ [25].

Substituting flour with WBPI within 30 min of dough development did not lead to significant changes in extensographic parameters compared to the reference flour. However, after 60 min from dough development, the energy increased by 16 cm^2^. This indicated that the length of the interval had an impact on the amount of energy when substituting up to 5% WBPI. As the substitution increased, the energy showed a decreasing trend. The dough with 15% WBPI substitution was evaluated as having the lowest energy resistance and stability, which was half that of the reference sample.

Dough resistance indicates the strength of gluten, meaning that higher resistance results in firmer gluten and a stronger and mechanically more resistant dough. The substitution with 5% WBPI showed a resistance approximately four times higher than that of the reference sample. Similar to the energy parameter, resistance also exhibited a decreasing trend with increasing WBPI substitution. The lowest gluten strength was observed in the case of 15% WBPI. The resulting dough was stiff but less mechanically resistant.

Dough extensibility indicated the elasticity of the dough. The reference sample showed extensibility around 180 mm, which varied depending on the maturation time. The optimal extensibility for unmodified flours used in baking ranges from 140 to 170 mm [2]. The sample with 5% WBPI substitution exhibited approximately half the extensibility compared to the reference sample. Samples with higher WBPI substitution were characterized as more brittle and less elastic. This phenomenon can be connected to the presence of dietary fiber in WBPI because it influences water redistribution in gluten and causes physical damage to the gluten matrix, which can result in reduced gluten visco-elasticity [26].

From the perspective of measuring dough energy, resistance, and extensibility, the dough properties improved with 5% WBPI substitution, where both energy and resistance were higher compared to the reference sample. This suggests that dough substituted with 5% WBPI exhibits greater mechanical resistance than the reference flour. On the other hand, samples with higher substitution levels (10% and 15%) showed deteriorating mechanical resistance.

In addition to resistance, the maximum of the extensographic curve also characterized the strength of gluten. Similar to the resistance parameter, the maximum parameter was the highest in the sample with 5% WBPI. A significant decrease in the maximum parameter occurred in the sample with 15% WBPI substitution. This decrease was twice as great as that for the reference flour (T530).

The ratio number indicated the ratio between resistance and extensibility, unlike the ratio number (max), which indicated the ratio between the maximum and extensibility. The most similar ratio number to the reference sample was found for the sample with 15% WBPI substitution. After 60 min of maturation, the samples showed a decreasing trend, except for the 5% WBPI substitution. The ratio number (max) was the lowest in the 15% WBPI sample, thus putting it the closest to the reference sample. The optimal value for the ratio number should be 2–2.5 although it could be higher for special flours. In the Czech Republic, the average extensographic ratio falls slightly below 2 for flours of medium quality [25]. From the overall assessment of the extensographic parameters of the prepared blends, it is evident that the dough with a 5% WBPI substitution appeared to be the best choice for further processing. As WBPI substitution increased, the extensographic quality decreased significantly.

Mixtures with whey protein had significantly different properties compared to the reference and mixtures with WBPI substitution. Substituting WPI flour led to a significant increase in dough energy compared to the reference sample. The increase in dough energy strengthened with increasing time interval from dough development. The high dough energy value with 5% WPI substitution indicates that the dough is not sensitive to processing conditions and will not soften rapidly during maturation and proofing.

The dough resistance value of the mixture with 5% WPI did not show the same positive trend as the energy mentioned earlier in the case of WBPI substitution. The extensibility of the dough with 5% WPI did not show a significant difference compared to the reference sample.

For energy, resistance, and extensibility measurements, the dough properties improved with 5% WPI, as both energy and extensibility were higher compared to the reference sample. The resistance value was not significantly lower than that of the reference, indicating that the dough fortified with 5% WPI exhibits similar mechanical strength to the reference sample.

The maximum dough value with 5% WPI showed the highest value after a 60 min resting period compared to the reference. However, it was significantly lower than the maximum values for the 5% WBPI sample. The ratio number did not reveal a significant difference between it and the reference.

### 3.4. Influence of WBPI and WPI Fortification of Sensory Properties of Pastry

The perception of sensory quality is a complex process related to the perception of appearance, taste, aroma, and texture. Sensory analysis was conducted on seven prepared samples of wheat baguettes that contained different amounts and types of protein.

Twenty independent assessors participated in the sensory analysis and evaluated the samples using a sensory questionnaire. The analysis took place in a specialized sensory laboratory following the procedure described in Section 2.2.6. The results of the baking experiments and their sensory analyses are divided into two sections—textural properties and aroma and taste properties. The median values for the textural and the aroma and taste parameters are presented in Table 9.

#### 3.4.1. Textural Properties of Fortified Pastry

The fundamental parameter of bakery products is their texture, which includes structure, surface, hardness, and chewiness. Among the most important textural properties of baguettes are high crispiness and firmness. Figure 2 provides an overview of individual textural parameters and the influence of the addition and type of protein isolate.

Structure analysis entails evaluating the strength, crumbliness, and cohesion of the fortified bakery product. From Figure 2, it is evident that the substitution of a protein isolate influenced the product’s structure, the primary parameter for describing textural properties. The substitution of wheat flour with WPI negatively affected the structural parameter, causing assessors to reduce the score by 1 unit compared to the reference sample. The decreased score was most likely caused by the fact that whey protein is highly hygroscopic. The substitution of WPI flour inhibited the structure of the gluten network, resulting in a decreased gluten concentration and competition among gluten, starch, and whey protein for water binding, leading to deterioration in the product’s cohesion. This effect was also observed during dough processing. Dough with a higher addition of whey protein was very watery and difficult to process, confirming the hygroscopic effect and weakened gluten network [27].

Sensory analysis did not confirm differences in the structure of the protein-enriched baguettes compared to the reference. However, a more in-depth visual examination of the structure of the baguettes in cross-section revealed that as the concentration of protein isolate from bran increased in the sample, there was a reduction in the expansion of gas cells during the baking phase (Figure 3). This hypothesis was supported by a study conducted by Alzuwaid et al. (2020) [28].

Surface is the second parameter for describing textural properties. No significant influence of either WPI or WBPI was observed on the surface of the baguettes. However, the substitution of WBPI flour resulted in an improvement in the surface rating by 1 unit compared to the reference sample. The higher score for the surface parameter in the WBPI-fortified baguettes was likely associated with the alteration of its color to lightly browned. This color change could have evoked a healthier consumer perception of whole-grain bread, which was reflected in increasing market demand for such products [27].

In evaluating the hardness parameter, a Kruskal–Wallis ANOVA test revealed that fortified baguettes differed significantly with a probability of *p* = 0.0001. The highest WBPI substitution led to a significant increase in hardness compared to the reference sample. From Figure 2, it can be seen that as the concentration of protein isolate from bran in the dough increased, so did the hardness of the sample. Conversely, in bread with whey protein, no specific correlation was observed between the amount of WPI and the hardness value. The fluctuation of values and the ambiguous trend in dough fortification with whey protein cannot be reliably explained based on the data, meaning that further detailed studies on this phenomenon are necessary.

Chewiness was the last evaluated textural characteristic of the fortified baguettes. A significant difference was observed for the fortified baguette with 15% WBPI (Figure 2). Compared to the reference value, an increase in chewiness (the product was less chewable) and a 1-unit score increase was observed. This result contradicted the conclusion of the study conducted by Alzuwaid et al. (2020), where the substitution of 5% WBPI flour increased chewiness (the product was less chewable). No significant influence of whey protein on the chewiness parameter was observed [28].

Regarding textural properties, the baguette with 15% WBPI flour substitution received the highest rating, while the baguette with 5% whey protein substitution received the lowest. The reference sample fell between the two. A correlation analysis revealed that none of the textural parameters correlated with the overall evaluation of the sample. Therefore, it could be concluded that textural properties do not have a significant impact on the overall evaluation of the product.

#### 3.4.2. Flavor and Aroma Properties of Fortified Pastry

Sensory analysis of taste and aroma is essential for assessing the properties of bakery samples. The taste parameter is considered to be a purely subjective evaluation by specific consumers. The formation of aromatic and flavor compounds depends on the conditions of yeast fermentation and its duration. Longer fermentation leads to products with a richer taste although bread is typically more acidic [29]. The formation of other aroma-active compounds occurred during the Maillard reaction. Figure 4 depicts the evaluation of aroma and taste parameters in bakery samples with different types and amounts of protein isolate.

The aroma of fortified baguettes is the first parameter, and to evaluate the differences among the samples, a Kruskal–Wallis ANOVA test was applied. Based on this analysis, fortified baguettes differed significantly at a significance level of *p* = 0.0003. It was found that the substitution of flour with a protein isolate affected the aroma. The highest decrease in score was observed for the baguette enriched with 15% WBPI, which experienced a decrease of 2.5 units compared to the reference sample. From Figure 4, it can be inferred that the pleasantness of the aroma decreased with higher substitutions of WBPI, intensifying the aroma of cereals, which was less pleasant for the assessors. Conversely, the addition of 5% whey protein resulted in an improvement of 1 unit in aroma compared to the reference sample. With a higher addition of WPI, the median value decreased to 4, which was half a unit lower than for the reference sample but not as significant. In the sample with 5% whey protein, the assessors most commonly described the aroma as pleasantly buttery or nutty.

Taste was another parameter evaluated, and from Figure 4 it is evident that independent assessors detected significant differences among the enriched wheat baguettes. Generally, taste acceptability decreased with higher protein substitutions. The sample with 15% WBPI experienced a decrease of 2 units compared to the reference sample, and the assessors detected a pronounced grainy taste. Similarly, WPI also negatively affected taste. The pastry with 15% WPI received the lowest rating, with the most commonly identified taste being buttery, which was less typical for bakery products.

Bitterness was another evaluated parameter among the samples where a statistically significant difference was identified as *p* = 0.00001. From Figure 4, it is apparent that only WBPI influenced the bitterness. In the 15% WBPI sample, bitterness increased by 2 units compared to the reference bread sample. The degree of bitterness increased proportionally with protein addition. The development of an undesired bitter-to-sour taste in the fortified bread could have been attributed to several probable causes. Wheat bran is rich in ferulic acid and tannins, which can impart a bitter taste, which in practice is masked by the addition of sweeteners (honey) or salt. Increased bitterness could also have been caused by certain amino acids: tryptophan, threonine, valine, phenylalanine, methionine, lysine, histidine, cysteine and arginine can cause bitterness when present in high concentrations. The increased attribute of bitterness with the addition of wheat bran was confirmed by the study conducted by Heiniö et al. [30], where the addition of bran layers to bread increased bitterness. Based on a correlation analysis, it was found that the bitterness parameter negatively correlated with the overall evaluation parameter (correlation −0.488). Based on the correlation coefficients, the lower the bitterness of the fortified bread, the better the overall evaluation [30].

For the saltiness parameter, neither WPI nor WBPI had a significant influence on the baguettes. The highest saltiness was exhibited by the reference baguette and the baguette with a 5% WBPI substitution. The saltiness of the fortified bread had a weakly positive correlation with the overall impression of the bread (correlation = 0.178).

The overall evaluation was the final parameter in the sensory analysis of the bread samples. It was found that the fortified baguettes differed significantly (*p* = 0.002) and that the reference baguette was rated as the most acceptable. Among the samples enriched with protein isolates, the baguettes with 5% WBPI and WPI exhibited the highest ratings with the same score. Generally, the overall impression decreased with higher isolate substitutions. The bitterness parameter had the most significant impact: as it increased, the overall acceptability of the bread significantly decreased. The samples made with dough where 15% of the flour had been replaced with WBPI were rated as almost unfit for consumption.

## 4. Conclusions

Farinographic measurements revealed that the substitution of flour with protein isolates significantly influenced both physiochemical and sensory properties of the dough and pastry. Enhancing the flour with WBPI resulted in a slight increase in flour water binding and an elevated degree of softening (12 min). However, other farinographic parameters indicated a reduction in dough quality. Conversely, the substitution of flour with WPI led to prolonged dough development time, enhanced dough stability, and improved farinographic quality.

Extensographic measurements demonstrated that WBPI-substituted dough exhibited comparable energy properties to the reference flour. The substitution of 5% WBPI flour resulted in improved energy and dough resistance, suggesting heightened mechanical strength compared to the reference sample. Nevertheless, as the level of WBPI flour substitution increased, extensographic parameters gradually declined, and there was no further enhancement in the dough’s mechanical properties. Samples with 5% WPI substitution exhibited superior mechanical properties compared to the reference sample.

The sensory analysis of the fortified baguettes revealed a significant impact of flour fortification by WPI and WBPI on the sensory properties of the pastry. The aroma was affected by the addition of protein isolates. Higher additions of WBPI resulted in less pleasant cereal-like aromas. Taste acceptability also decreased with higher protein additions. A pronounced grainy taste (both isolates) and buttery taste (WPI) were less typical for bakery products. A fundamental reduction in overall acceptance scores was observed in pastry with a high WBPI substitution level. A correlation analysis proved that this phenomenon was connected with bitterness. The bitter taste was probably caused by the presence of small peptides, ferulic acid and tannins. Future work should focus on the sensory masking of this defect.

The substitution of WBPI and WPI for flour represented a suitable approach for preparing high-protein flour mixes, which can be used in the production of high-protein pastry. WBPI is a promising dietary protein produced from widely available and sustainable material, and could be included among common protein isolates like soy and WPI.

## Figures and Tables

**Figure 1 foods-12-02635-f001:**
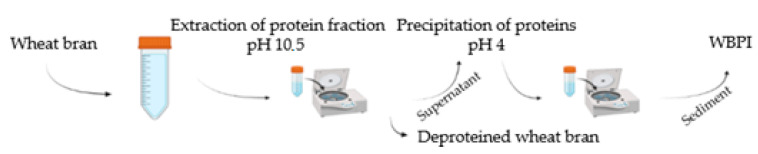
Isolation of WBPI.

**Figure 2 foods-12-02635-f002:**
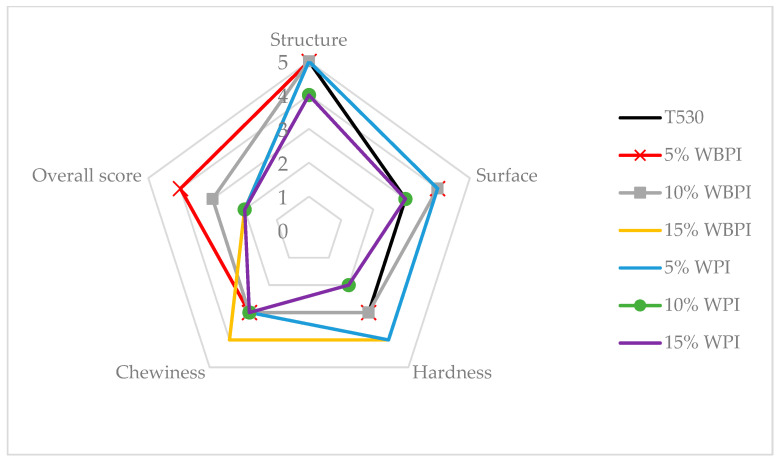
Textural parameters of pastry samples.

**Figure 3 foods-12-02635-f003:**
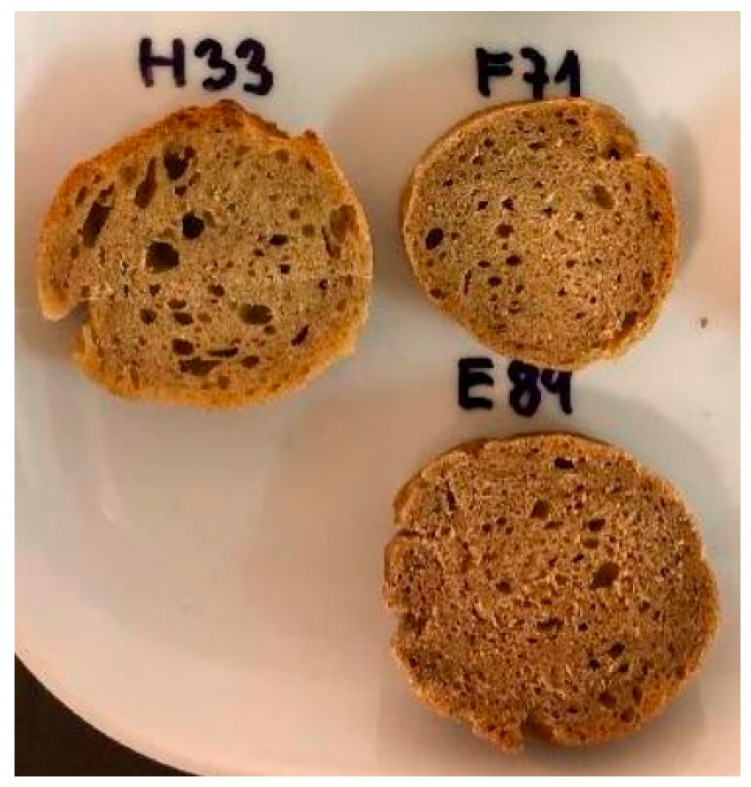
Gas cells of pastry samples: H33—5% WBPI; F71—10% WBPI; E84—15% WBPI.

**Figure 4 foods-12-02635-f004:**
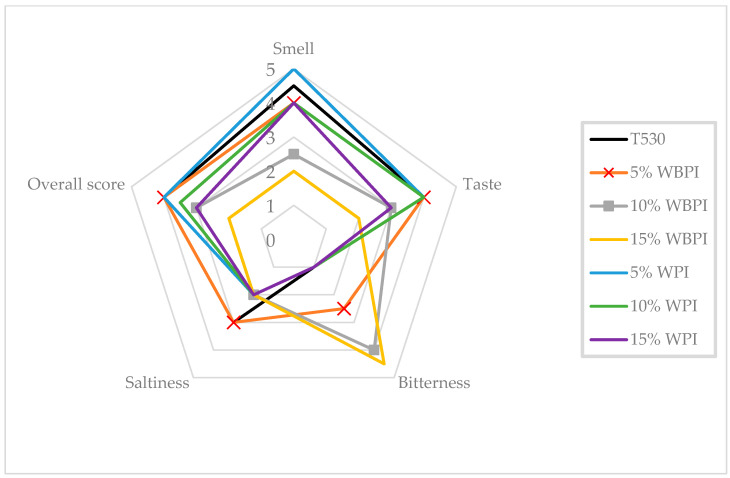
Flavor parameters of pastry samples.

**Table 1 foods-12-02635-t001:** Composition of flour mixtures.

	Flour Content (g)	Isolate Content (g)
T530	92.5	0
5% WBPI	87.9	4.6
10% WBPI	83.3	9.3
15% WBPI	78.6	13.9
5% WPI	87.9	4.6
10% WPI	83.3	9.3
15% WPI	78.6	13.9

**Table 2 foods-12-02635-t002:** Extensograph settings.

Speed of balling unit (min^−1^)	83.0 ± 3.0
Speed of dough roll (min^−1^)	15.0 ± 1.0
Speed of stretching hook (mm·s^−1^)	14.5 ± 0.5
Extensographic force (mN·EJ^−1^)	12.3 ± 0.3

**Table 3 foods-12-02635-t003:** Optimized water addition for extensographic measurements.

	Water Volume (mL)
T530	54.6
5% WBPI	52.6
10% WBPI	54.9
15% WBPI	57.7
5% WPI	47.9

**Table 4 foods-12-02635-t004:** Raw material for pastry production.

	Water (g)	Flour (g)	Isolate (g)	Yeast (g)	Salt (g)
T530	64	92.5	0	1.2	2.4
5% WBPI		87.9	4.6		
10% WBPI		83.3	9.3		
15% WBPI		78.6	13.9		
5% WPI		87.9	4.6		
10% WPI		83.3	9.3		
15% WPI		78.6	13.9		

**Table 5 foods-12-02635-t005:** Protein content and amino acid composition of protein isolates.

Amino Acid Content (%)		WBPI	WPI
	Met	1.2	1.6
	Met	1.2	1.6
	Lys	2.7	6.8
	Thr	2.2	5.4
	Asp	6.2	8.5
	Ser	3.2	4.9
	Glu	13.6	13.7
	Gly	3.5	1.5
	Ala	3.1	3.8
	Tyr	2.5	2.2
	Val	3.1	4.6
	Phe	2.8	2.5
	Ile	1.9	4.8
	Leu	4.6	8.1
	His	2.2	1.3
	Arg	5.2	1.9
	Cys	1.1	1.6
	Pro	6.4	4.9
	Trp	0.6	1.3
**Protein content (%)**		67.6	76.9

**Table 6 foods-12-02635-t006:** Characteristics of flour and high protein blends.

	Protein Content (%)	Water Content (%)	Wet Gluten Content (%)	Sedimentation Volume (mL)
T530	13.8	10.2	38.6	21.0
5% WBPI	16.5	9.8	33.5	25.0
10% WBPI	19.2	9.4	31.2	28.0
15% WBPI	21.9	9.1	28.3	42.0
5% WPI	16.9	10.0	33.1	23.0
10% WPI	20.1	9.8	30.5	20.0
15% WPI	23.3	9.7	19.6	17.0

**Table 7 foods-12-02635-t007:** Farinographic characteristics of flour mixtures.

	T530	5% WBPI	5% WPI
Water absorption [%]	55.8	56.4	52.6
Development time [min]	1.8	1.7	6.8
Stability [min]	3.8	2.6	9.8
Softening degree (10) [FU]	72.0	73.0	20.0
Softening degree (12) [FU]	94.0	75.0	74.0
Farinographic quality number	35.0	30.0	113.0

**Table 9 foods-12-02635-t009:** Sensory descriptor scores.

	Structure	Surface	Hardness	Chewiness	Smell	Taste	Bitterness	Saltiness	Overall Score
Reference	5.0	3.0	3.0	3.0	4.5	4.0	1.0	3.0	4.0
WBPI 5	5.0	4.0	3.0	3.0	4.0	4.0	2.5	3.0	4.0
WBPI 10	5.0	4.0	3.0	3.0	2.5	3.0	4.0	2.0	3.0
WBPI 15	5.0	4.0	4.0	4.0	2.0	2.0	4.5	2.0	2.0
WPI 5	5.0	4.0	4.0	3.0	5.0	4.0	1.0	2.0	4.0
WPI 10	4.0	3.0	2.0	3.0	4.0	4.0	1.0	2.0	3.5
WPI 15	4.0	3.0	2.0	3.0	4.0	3.0	1.0	2.0	3.0
ANOVA P	0.0474	0.5985	0.0001	0.0024	0.0030	0.0004	0.0001	0.5763	0.0002

## Data Availability

The data presented in this study are available on request from the corresponding author.

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
