# Peer review of "Physiochemical and Sensory Properties of Bread Fortified with Wheat Bran and Whey Protein Isolates"

_foods, 2023, doi:10.3390/foods12132635_

Round 1

Reviewer 1 Report

The study aims to investigate the effects of wheat bran protein isolates and whey protein isolates on physiochemical and sensory properties of pastry. The statement appears to be confusing in prepared samples or analyzed samples, and the statement is unclear and incomplete in the result section. And what the most importance is that I did not find any novel point in this study. 

This manuscript is full of errors and irregularities in English. I recommend checking it carefully before submitting it again. 

Author Response

Dear reviewer,

thank you for taking a time to review our manuscript. Your comments were very helpful to improve quality of our manuscript. Your comments and recomendations have been taken into account in the revised version of the article and are commented on point by point in the attached file. 

Thank you very much

sincerely

Jaromír PoÅ™ízka and author team

Reviewer 2 Report

The manuscript seems interesting and needs to be revised considerably.

English should be checked again.

Author Response

(The authors gave the same response as above.)

Round 2

Reviewer 1 Report

Please check the minor editing of English language. 

Accept in present form.

Reviewer 2 Report

The manuscript can be accepted.

Improved.